# The Impact of the War in Ukraine on the Epidemiological Situation of Tuberculosis in Europe

**DOI:** 10.3390/jcm12206554

**Published:** 2023-10-16

**Authors:** Natalia Anna Wilczek, Anna Brzyska, Julia Bogucka, Wiktoria Ewa Sielwanowska, Monika Żybowska, Halina Piecewicz-Szczęsna, Agata Smoleń

**Affiliations:** The Chair and Department of Epidemiology and Clinical Research Methodology, Medical University of Lublin, 20-081 Lublin, Poland; a.brzyska09@gmail.com (A.B.); bogucka_julia@gazeta.pl (J.B.); wiktoria.sielwanowska@gmail.com (W.E.S.); zybowska.m@gmail.com (M.Ż.); halina.piecewicz-szczesna@umlub.pl (H.P.-S.); agata.smolen@umlub.pl (A.S.)

**Keywords:** tuberculosis, epidemiology, refugees, war in Ukraine, migration

## Abstract

Ukraine is at the forefront of one of the largest outbreaks of tuberculosis (TB) in Europe, including multidrug-resistant tuberculosis. Due to the ongoing armed conflict in this area, there is a significant number of refugees from Ukraine to other European countries. The aim of the study is to emphasize the essence of the problem of the increased incidence of tuberculosis, taking into account the impact of the armed conflict in Ukraine, resulting in the intensification of migration movements. A descriptive epidemiological method was used as the research method. The material was collected by analyzing source statistical data from WHO and statistical yearbooks of selected European countries. Particularly, Ukrainian refugees in Poland, Germany, the Czech Republic, and Slovakia are at higher risk of contracting tuberculosis due to factors like migration stress, poor nutrition, and comorbidities. Epidemiological data from these countries show a rise in tuberculosis cases among foreigners, emphasizing the need for European specialists to be more vigilant in this unique situation, focusing on refugees and other vulnerable populations. More research and collaborative efforts are essential to closely monitor and prevent the severe outcomes of tuberculosis transmission.

## 1. Introduction

The Russian invasion of Ukraine, which began on 24 February 2022, has contributed to a massive humanitarian crisis and disruption of health care [1]. As a result of that invasion, many people were forced to leave their former possessions, which contributed to large-scale migration movements, both within the country and to neighboring regions [2]. From the data published by Koroutchev in mid-March 2023, results of movements of the Ukrainian population, especially to Central European countries such as Poland, Germany, the Czech Republic, Slovakia, Moldova, and Hungary, were observed [3]. The phenomenon is worrying because according to the WHO report, Ukraine is currently the country with the fourth highest incidence of TB in Europe, and what is more, it also ranks 5th worldwide in the number of registered cases of drug-resistant tuberculosis [4]. Therefore, it has been classified by the WHO as a high-priority country (HPC), i.e., one of 18 countries that are responsible for slightly more than 80% of cases and almost 90% of deaths due to tuberculosis in Europe [5]. The tuberculosis epidemic is driven by many factors, which in the aspect of the ongoing war even more strengthen their importance. WHO, in the report from 2022, lists the most dangerous and most common risk factors for tuberculosis, infection with the immunodeficiency virus, malnutrition, alcohol consumption, smoking, or diabetes. Armed conflicts both in Ukraine and in other parts of the world, as well as the global food and energy crisis, may strengthen the impact of some determinants of TB, e.g., economic decline and malnutrition [6]. The situation in which Ukrainian citizens find themselves is struggling with difficult access to medical care, large concentrations of people staying in shelters with insufficient ventilation, and accompanying stress, which also favors the spread of tuberculosis [7]. Even before the outbreak of the war, due to the alarming statistics of TB cases, innovations and reforms were implemented to combat the problem. The ongoing war poses a significant threat to the progress achieved so far in this area and, in the context of the conflict under consideration, is a significant challenge for both Ukraine and the countries supporting it [8]. All the consequences are currently not only borne by Ukraine but also due to the sanctions imposed on the flow of refugees. This problem is imposed on various sectors, both economic and health, in Europe and even the whole world.

## 2. Materials and Methods

### 2.1. Study Design and Objective

A descriptive epidemiological method was used as the research method, which was carried out thanks to a collection of epidemiological data on the incidence of tuberculosis in selected European countries from the period before the war in Ukraine, covering the years 2018–2021 and 2022, when the Russian-Ukrainian conflict broke out on February 24. The aim of the study is to show changes taking place in the incidence of tuberculosis in Europe in the face of the ongoing war in Ukraine and to increase the vigilance of medical professionals towards people at risk of developing tuberculosis. In particular, we want to highlight the situation of a significant number of refugees from Ukraine who moved to Europe and epidemiological changes that were observed in the context of tuberculosis in some European countries that accepted large numbers of war refugees from Ukraine.

### 2.2. Data Collection

The material was collected by analyzing source statistical data from WHO and statistical yearbooks of some European countries, such as Ukraine, Poland, Germany, the Czech Republic, and Slovakia. Nationally representative data on the incidence of tuberculosis is derived from a range of sources broadly defined as national administrative data. National administrative data are those coming from national systems, including registration of infectious diseases and statistics systems and national health management information systems. The presentation of data in charts was prepared using Microsoft Excel (2016). It is worth pointing out that the latest epidemiological data from Ukraine are statistical data without occupied territory and without territory where hostilities are taking place. Detailed sources of the analyzed data are provided below for a given country.

(1)Poland

Gruźlica i Choroby Układu Oddechowego w Polsce w 2018 r. red. Maria Korzeniewska-Koseła. Instytut Gruźlicy i Choro Płuc, Warszawa 2019Gruźlica i Choroby Układu Oddechowego w Polsce w 2019 r. red. Maria Korzeniewska-Koseła. Instytut Gruźlicy i Chorób Płuc, Warszawa 2020Stan Sanitarny Kraju w 2022 Roku—Wojewódzka Stacja Sanitarno-Epidemiologiczna w Opolu. https://www.gov.pl/web/wsse-opole/stan-sanitarny-kraju-w-2022-roku (accessed on 9 August 2023)Gruźlica i Choroby Układu Oddechowego w Polsce w 2022 r. red. Maria Korzeniewska-Koseła. Instytut Gruźlicy i Chorób Płuc, Warszawa 2023Światowy Dzień Gruźlicy 2023—Główny Inspektorat Sanitarny. https://www.gov.pl/web/gis/swiatowy-dzien-gruzlicy-2023 (accessed on 9 August 2023)Światowy Dzień Gruźlicy w Tym Roku pod Hasłem Tak! Możemy Wyleczyć Gruźlicę. https://szczepienia.pzh.gov.pl/swiatowy-dzien-gruzlicy-w-tym-roku-pod-haslem-tak-mozemy-zwalczyc-gruzlice/ (accessed on 24 March 2023)

(2)Ukraine

Cтaтиcтикa з TБ. Цeнтp Гpoмaдcькoгo Здopoв’я. https://phc.org.ua/kontrol-zakhvoryuvan/tuberkuloz/statistika-z-tb (accessed on 11 August 2023)WHO: “WHO Information Note on Ensuring Continuity of Essential Tuberculosis Services for People with or at Risk of the Disease within Ukraine and in Refugee-Hosting Countries” 2022. https://www.who.int/publications/i/item/9789240050747 (accessed on 22 May 2022)Cпeцiaлicт/кa з Yпpaвлiння Пpoєктaми (TB and Infectious Disease). Гpoмaдcький Пpocтip. https://www.prostir.ua/?jobs=spetsialistka-z-upravlinnya-projektamy-tb-and-infectious-disease (accessed on 1 August 2023)Пpoблeмa Тyбepкyльoзy пiд чac Вiйни: Щo Пoтpiбнo Знaти тa Кyди Звepтaтиcя зa Дoпoмoгoю. Гpoмaдcький Пpocтip. https://www.prostir.ua/?news=problema-tuberkulozu-pid-chas-vijny-scho-potribno-znaty-ta-kudy-zvertatysya-za-dopomohoyu (accessed 7 August 2023).Як Вiйнa Пoгipшyє Eпiдeмiчнy Cитyaцiю в Укpaїнi. Deutsche Welle (DW) 2022. https://www.dw.com/uk/tuberkuloz-vil-ta-kholera-yaki-ryzyky-nese-viina-dlia-zdorovia-ukraintsiv/a-62092825 (accessed on 7 August 2023).At the Polish Border, Tens of Thousands of Ukrainian Refugees https://www.nytimes.com/2022/02/25/world/europe/ukrainian-refugees-poland.html (accessed on 25 February 2023)Зbit зa Peзульtatamи Дocлiджehhя “Бap’єpи Bctahobлehhя Дiaгhoзу ta Лikуbahhя Tубepkульoзу у Cлiдчиx Iзoляtopax, Уctahobax Bиkohahhя Пokapahь ta Cпeцiaлiзobahиx Tубepkульoзhиx Лikaphяx ЦOЗ ДKBC Уkpaïhи” https://phc.org.ua/sites/default/files/users/user90/ZvitTB_Full.pdf (accessed on 9 September 2023)Цeнтp Мeдcтaтиcтики—зa 2021. http://medstat.gov.ua/ukr/MMXXI.html?fbclid=IwAR2lEKBio9iGyLM2WKkxUpkXcjnI2aoGo7r32aZU_2-izhCkSA72NWfqOCw (accessed on 9 September 2023).Cтaтиcтикa з TБ. Цeнтp Гpoмaдcькoгo Здopoв’я. https://phc.org.ua/kontrol-zakhvoryuvan/tuberkuloz/statistika-z-tb (accessed on 11 August 2023)Ukrainian Refugee Cost by Country 2022. Statista. https://www.statista.com/statistics/1312602/ukrainian-refugee-cost-by-country/ (assessed on 18 July 2023)

(3)Germany

Geflüchtete aus der Ukraine in Deutschland. Mediendienst Integration. https://mediendienst-integration.de/artikel/gefluechtete-aus-der-ukraine-in-deutschland.html (accessed on 5 January 2023)Epidemiologisches Bulletin. 2023. https://www.rki.de/DE/Content/Infekt/EpidBull/Archiv/2023/Ausgaben/11_23.pdf?__blob=publicationFile (accessed on 16 March 2023)

(4)Czech Republic

Čechová, Š. Situace s Výskytem TBC Zůstává v ČR Příznivá, Ukazují Data. SZÚ. Oficiální Web Státního Zdravotního Ústavu v Praze. https://szu.cz/aktuality/situace-s-vyskytem-tbc-zustava-v-cr-prizniva-ukazuji-data/ (accessed on 22 february 2023)

(5)Slovakia

Tuberculosis Surveillance and Monitoring in Europe 2020—Data 2022 https://www.ecdc.europa.eu/en/publications-data/tuberculosis-surveillance-and-monitoring-europe-2022-2020-data (accessed on 24 March 2022)Tuberculosis Surveillance and Monitoring in Europe 2018—Data 2020 https://www.ecdc.europa.eu/en/publications-data/tuberculosis-surveillance-and-monitoring-europe-2020-2018-data (accessed on 24 March 2020)Problematika Tuberkulózy na Slovensku Aktuálne—NUTPCHaHCH Vyšné Hágy, Národný Register Tuberkulózy NCZI https://www.hagy.sk/narodny-register-tbc/analyza-situacie-tbc-na-slovensku/?fbclid=IwAR3u07K3SL9JGypv44W1Y9mnDRvY5VJDYalCQsS2CWwvz2jL2h3vdfmfbGY (accessed on 24 August 2023)Analýza Epidemiologickej Situácie a Činnosti Odborov Epidemiológie v Slovenskej Republike za Rok 2021 https://www.epis.sk/InformacnaCast/Publikacie/VyrocneSpravy/Files/VS_SR_2021.aspx?fbclid=IwAR0UvNIEI2gmxLQ-2S1WdpUC0vZcMuXo82l3R-QJwX-hZMA8WwoE-UlU69s (accessed on 24 August 2023)Tuberculosis Surveillance and Monitoring in Europe 2021—Data 2023 https://www.ecdc.europa.eu/sites/default/files/documents/tuberculosis-surveillance-monitoring-2023.pdf?fbclid=IwAR1DaxBsZ6nuYXwQBg2ilXVyn_o1ZdFs2_pa__Vlais7Ico39K6MKOS_AHw

## 3. Results

### 3.1. Tuberculosis as a Relevant Challenge for Ukraine

From 2015 to 2018, a decrease in the total number of tuberculosis cases could be observed in Ukraine. In 2018, 25% fewer cases were recorded compared to 2015. This decrease was mainly due to greater efforts made by the state authorities to fight tuberculosis. 2020 turned out to be the year with the lowest number of cases in years. This was most likely the result of the outbreak of the COVID-19 pandemic and limited access to health services, so the number of cases is underestimated. The next two years were associated with an increase in reported cases of tuberculosis. The Public Health Center in Ukraine reported that in 2022, 18,510 cases of tuberculosis were registered in the country, which is a significant increase of over 2.5% compared to 2021 (Figure 1).

Ukraine is also a country where multidrug-resistant tuberculosis (MDR) is very common, which is characterized by resistance to treatment with commonly used and strongest anti-tuberculosis drugs, i.e., isoniazid and rifampicin. Drug resistance in tuberculosis can have various causes. Improper use of drugs, especially irregular use and taking breaks between doses, contributes to the rapid development of drug resistance. In the international arena, it is Ukraine that is at the forefront of one of the largest outbreaks of this disease. Every year, an estimated 32,000 people become infected with active tuberculosis. The drug-resistant form occurs in about 33% of people with tuberculosis, which is 30 times more frequent than in Poland. In 2020, 4257 cases of drug-resistant tuberculosis (MDR-TB) were registered in Ukraine. In comparison, only 566 cases have been reported in the European Union/European Economic Area. Over the three years from 23 February 2019 to 23 February 2022, there was a gradual decrease in the average number of hospitalized adult tuberculosis patients (233.6 ± 152.9 vs. 100.7 ± 61.9; *p* < 0.001). During the period from February 2022 to August 2022, the number of hospitalized adult tuberculosis patients remained relatively constant. Another problem is that the treatment of multidrug-resistant tuberculosis (MDR-TB) in Ukraine is unsatisfactory. The treatment success rate remains low at around 50% in 2021.

### 3.2. Situation of TB Cases in Selected European Countries after the War in Ukraine Outbreak

An important aspect that needs attention is the presence of a large number of refugees who move from Ukraine to other countries. The resulting migration stress, the presence of comorbidities, and inadequate nutrition put expatriates at a significantly higher risk of contracting tuberculosis. By the end of the first quarter of 2022, almost 6 million people left the territory of Ukraine, seeking refuge in other countries. In the following months of 2023, the number of refugees from Ukraine who came to Poland is estimated at approximately 700,000. The largest number of people from Ukraine resettled to neighboring countries beyond their western border. The leading countries that have accepted the largest number of Ukrainian citizens are Poland, Germany, the Czech Republic, and Slovakia, which is perfectly illustrated on the map below (Figure 2). Therefore, below, we have developed the epidemiology of tuberculosis in these countries, including the variability of the number of cases among foreigners.

#### 3.2.1. Poland

Specifying the epidemiological data of Poland, the incidence of tuberculosis in 2020 was 36.7% lower compared to 2019. In turn, in 2021, 3704 cases of tuberculosis were registered in Poland, which is 316 more cases than the previous year and 3838 fewer cases compared to 2012. In addition, the incidence of all forms of tuberculosis in 2021 was 9.7 per 100,000 population, which was 10.2% higher than in 2020 (8.8) and 50.5% lower than in 2012, when the incidence rate was 19.6 per 100,000 population. In Poland, over the last five years, the situation of tuberculosis cases has been changing dynamically, and the growing share of foreigners in the statistics of cases attracts attention, which we would like to present on the diagram below (Figure 3). In 2018, 5487 new cases of tuberculosis were recorded, including 97 foreigners. A year later, the number of new cases decreased slightly (5321). However, an increase in the share of cases among foreigners was recorded (122). In the next two years, due to the limited access to healthcare caused by the outbreak of the COVID-19 pandemic, a decrease in documented cases of tuberculosis was observed, i.e., 3388 in 2020 (including 116 foreigners) and 3704 in 2021 (including 132 foreigners). In turn, data from 2022 report that as many as 4205 cases were registered at that time. Whereof 269 cases were among foreigners, which means a doubling of the number of cases among this group of people compared to 2021, and the majority of cases, i.e., 175, were citizens of Ukraine (Figure 4). The latest reports from the Chief Sanitary Inspectorate in Poland suggest that in the first half of 2023, 1566 cases of TB were confirmed in Poland.

#### 3.2.2. Germany

Despite the overall decrease in tuberculosis (TB) incidence between 2018 and 2022 in Germany, a concerning trend is the rising proportion of foreign nationals among TB patients. In 2022 alone, Germany saw an influx of nearly one million individuals of Ukrainian origin. In light of the ongoing war and intensified migratory movements, an increase in tuberculosis (TB) cases among foreigners has been observed within the territory of Germany. This trend is particularly pronounced among individuals immigrating from Ukraine. In 2022, there were 2847 reported cases of TB among foreigners, with 262 of these cases among the Ukrainian population—an increase of 236 cases compared to the previous year. The reported TB case numbers for the years 2020–2022 may be slightly underestimated due to the ongoing COVID-19 pandemic and limited access to healthcare. Therefore, the actual number of cases is likely higher (Figure 5).

#### 3.2.3. The Czech Republic

Similar to Poland and Germany, in the Czech Republic, there has been an observed increase in tuberculosis (TB) cases among foreigners during the period of 2018–2022. In 2022, the highest number of cases among individuals from outside the country was recorded in the past 5 years—156 cases, including a significant 87 cases among the Ukrainian population, which is over twice as many as the previous year (Figure 6). Despite the relatively low overall number of cases within the Czech territory, the alarming aspect is the rapid growth of reported tuberculosis cases among foreigners.

#### 3.2.4. Slovakia

In 2022, the number of tuberculosis cases among foreigners was the highest in the period 2018–2022 (Figure 7). People from Ukraine accounted for 70.6% (12 cases) of all foreigners’ cases. There is a noticeable increase here compared to 2021, where people from Ukraine accounted for 28.6% (2 cases) of all cases of foreigners (Figure 8). In 2018, foreigners accounted for 3.6% of all cases, 3.3% in 2019, 3.8% in 2020, and 5.1% in 2021.

## 4. Discussion

The aim of this analysis was to draw attention to the increasing incidence of tuberculosis, which is an important alarm signal to raise awareness of this disease. Especially nowadays, when the world is facing serious challenges regarding energy and food security, which the armed conflict in Ukraine has contributed to. The ongoing war has a multidimensional impact on the issue of public health in the world, which has already been weakened in recent years by the COVID-19 pandemic and ongoing climate change [9,10,11,12,13]. The armed conflict in Ukraine has a negative impact on many aspects related to the fight against tuberculosis. Given that this is a region in Europe where tuberculosis is a serious problem, especially with the level of MDR-TB that is also common there, military action damages the health system and could undermine the efforts made so far to eradicate that disease [5,10,11,14,15]. In addition, other factors that can significantly delay and worsen the fight against tuberculosis in Ukraine are other infectious diseases common in the country, while in other countries, these diseases have been largely eradicated or effective treatments exist [16]. This is a huge risk because each subsequent disease is a factor that overloads the capacity of healthcare systems, which results in poorer quality of services and often an inability to provide appropriate therapy and even detect the disease [17]. An example of such a disease is measles. In Ukraine, there was a large measles epidemic in the period 2017–2020, where more than 115,000 cases of the disease were registered [16]. Furthermore, this is also due to the country’s low vaccination rate. Contributing to this is the growing distrust in the area of medical care caused by misinformation and the resulting lack of awareness of the dire consequences of infectious diseases for which vaccines are available. An important proof is the level of vaccination coverage of the Ukrainian population due to COVID-19, which was only 36%. Unfortunately, this problem has not gone away, and the ongoing armed conflict has also contributed to slowing down the polio vaccination campaign conducted by the Ukrainian Ministry of Health [18]. According to the WHO, the vaccination rate of a given population with BCG vaccination should reach 95%. In Ukraine in the previous year (2022), it was only 77% [19]. Even before the outbreak of the war, Ukraine led the way in terms of tuberculosis cases in Europe, including drug-resistant tuberculosis [4]. Due to the warfare taking place mainly in the east and south of the country, the Ministry of Health named the number of medical facilities destroyed as a result of the war: more than 1400 medical facilities were damaged, and 177 were completely devastated. According to the report of the Minister of Health of Ukraine, losses in the Ukrainian health sector are estimated at tens of billions of dollars. According to the calculations of experts from the World Bank and WHO, losses in the second half of September 2022 were estimated at about USD 26 billion. In addition to the existing damage, Russian soldiers removed drugs and valuable medical equipment from facilities such as hospitals, clinics, and laboratories [8]. In the situation in which Ukrainian citizens find themselves, not only people from risk groups are at risk, but also the specific conditions in which citizens find themselves contribute to the increase in the number of tuberculosis cases. It is well known that the weakening of the immune defense significantly contributes to the infection and development of tuberculosis. Long-term physical and mental stress, including the presence of increased levels of chronic stress, food shortages, and staying in large population centers, for example, in bomb shelters with limited ventilation, contribute to a decrease in immunity, which can potentially increase the spread and development tuberculosis [7,20,21]. For doctors who care for patients in regions where hostilities are underway, executing their job is risky and difficult. They often fear for their lives, and the specific working conditions they find themselves in make it difficult to help effectively [4]. It is believed that the percentage of the actual number of cases of tuberculosis in Ukraine has already exceeded 0.5% of the country’s population. There are concerns about further observation of the increase in the number of cases, bearing in mind the epidemiological data after World War I or World War II. Programs that increase awareness of the risks and consequences of TB infection by providing the public with simple and understandable messages on this subject are important [22]. Some people do not go to the general practitioner with disturbing symptoms because during martial law and the related resettlements to other parts of the country, fighting for their lives, the war puts other priorities in the foreground. Avoiding visits to the doctor also results from disinformation. Hence, it seems that the number of cases reported during the war is underestimated, if only for the reasons mentioned above. Only post-war statistics will show the real numbers of infections [19,23]. In addition, the threat of the risk of spreading infectious diseases also increases with the resettlement of Ukrainian citizens from the most vulnerable eastern regions to the west of the country [24]. The outbreak of war has significantly reduced the socio-economic level in the region, increased the level of malnutrition, and limited access to basic health care. Therefore, people migrating from these areas are at a higher risk of contracting tuberculosis [9,25,26,27]. It also constitutes a serious challenge to the progress of European countries in the fight against tuberculosis, especially in Poland, which, at the beginning of the war, received a huge number of refugees from that region at the very beginning of the war [16,17,21,28]. For a year, in the period from 24 February 2022 to 24 February 2023, the ongoing armed conflict caused by the Russians, slightly more than 10 million people crossed the Polish-Ukrainian border, mainly women with children and elderly people. Data from the beginning of 2023 indicate that about 2 million Ukrainians have settled in Poland [29]. The influx of refugees to Poland has caused many challenges for the Polish state in the healthcare sector. The data presented covering selected countries around the world in the period from the outbreak of the war to June 2023 suggest that Poland incurred the largest total social and health costs related to the reception of Ukrainian refugees. The burden of costs on the Polish healthcare system due to this situation is estimated at EUR 15.4 billion. The second largest financial outlay was Germany, where the cost of accepting refugees from Ukraine is estimated at EUR 13.9 billion [30]. Due to the need to react quickly to the needs of our neighbors who have experienced Russian aggression, the Government of the Republic of Poland passed a law on 12 March 2022 that ensures equal access for Ukrainian citizens to healthcare services, including reimbursement of medicines, vaccinations, and rehabilitation [12]. Also causing concern is the growing trend of tuberculosis cases among foreigners in recent years in countries such as Germany, the Czech Republic, and Slovakia. In Germany, the number of people of other origins suffering from tuberculosis is higher than among native-born citizens. Attention is also drawn to the growing share of Ukrainians among tuberculosis cases in mentioned countries in 2022 compared to previous years [31,32,33,34,35,36,37,38]. All these factors encourage the inclusion of all of the displaced people in the program of screening tests for tuberculosis diagnosis and to control the situation of the disease among foreigners [25]. Failure to take appropriate action can lead to the rapid spread of M. tuberculosis around the world, which is becoming a major epidemiological problem. In the aspect of the healthcare system of these European countries, it is also important that people coming from Ukraine with chronic and oncological diseases, often without full medical documentation or a language barrier, cause numerous difficulties in specialized care for these patients [39]. The current situation, which manifests itself in the inevitable and long-term increase in demand for a larger and more diverse population, prompts the necessary restructuring of the healthcare system in Poland to ensure optimal and effective access to health services [40]. Tuberculosis still remains a significant public health problem in the 21st century. Every effort should be made to make progress that can revolutionize the treatment, counteraction, and prevention of the undesirable effects of this disease. The WHO strategy to eliminate tuberculosis aims to reduce the number of deaths by 95%, as well as reduce morbidity by 90% by 2035. Furthermore, it includes a commitment to all United Nations (UN) member states to end the global TB epidemic by 2030. Therefore, the vigilance of specialists should be increased, especially in individuals presenting an increased risk of infection, without forgetting the ongoing armed conflict in Eastern Europe, the indirect impact of which may have fatal consequences in the context of progress in combating tuberculosis in view of the disease can take place in almost every area of the body, and its symptoms can imitate many disease entities [41,42]. Managing the crisis related to the increase in tuberculosis cases in Europe as a result of the influx of migrants from Ukraine during the war requires many actions at various levels, both nationally and internationally. It is particularly important to support systems for monitoring and identifying cases of tuberculosis not only among migrants but among the entire population. For this purpose, it is necessary to improve cooperation between public health facilities, migration centers, and international organizations. Other necessary interventions are an increase in the availability of tests for detecting tuberculosis and a systematized and publicly available system for rapid and accurate diagnosis and treatment of patients with tuberculosis. The introduction of isolation of people with active tuberculosis may also significantly reduce the spread of the disease. An important element is also improving public awareness of the threats posed by the increase in tuberculosis cases. Implementation of educational campaigns about tuberculosis, how it spreads, its symptoms, prevention, and available support among migrants and communities. Additionally, care should be taken to introduce protective vaccinations against tuberculosis among unvaccinated migrants arriving in European countries. Migrants should be provided with psychological support to facilitate their integration process, which may improve their access to health care and education and reduce the risk of spreading diseases. Finally, diplomatic actions are necessary throughout the international arena to end the armed conflict in Ukraine as quickly as possible, which is the main cause of migration and the health crisis. All these strategies should be implemented in a balanced way, respecting human rights and taking into account the needs of migrants and local communities. In the effective management of the health crisis, international cooperation as well as solidarity between European countries are extremely important [4,43,44].

### Strengths and Limitations

This article included sources of statistical data from selected European countries that hosted the largest number of Ukrainians after the war outbreak to quantify and study the prevalence of tuberculosis in their populations in countries in 2018–2022. The article raises a very important and contemporary problem, emphasizing the possibility of mutual influence between socio-political upheavals and health crises. However, the present analysis was not without limitations. Descriptive studies are useful for estimating the burden of disease, e.g., prevalence or incidence in a population. This information is useful only for initial problem identification resource planning. This helps generate hypotheses regarding the cause of the problem, which should then be verified using another, more complex design, e.g., analytical methods such as cohort or case-control study.

## 5. Conclusions

In 2022, when the war broke out, a noticeable increase was observed in the share of people born in Ukraine among tuberculosis cases in Poland, Germany, the Czech Republic, and Slovakia compared to previous years. Increased migration movements to European countries as a result of the armed conflict in Ukraine may potentially contribute to the increasing percentage of tuberculosis cases in these countries. There is a need to take further actions requiring multi-sectoral cooperation to monitor the spread of tuberculosis, as well as to continue educational and preventive programs to increase awareness of this disease in society. Doctors in Europe must also be more vigilant towards risk groups, especially refugees, given the background of this situation.

## Figures and Tables

**Figure 1 jcm-12-06554-f001:**
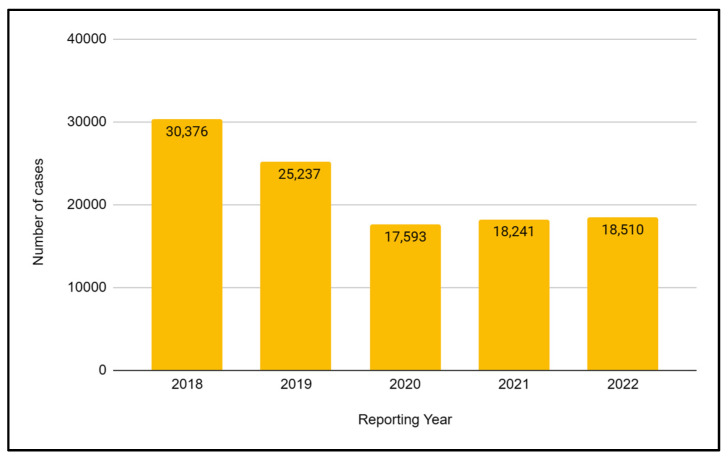
Number of tuberculosis (TB) cases in Ukraine in 2018–2022.

**Figure 2 jcm-12-06554-f002:**
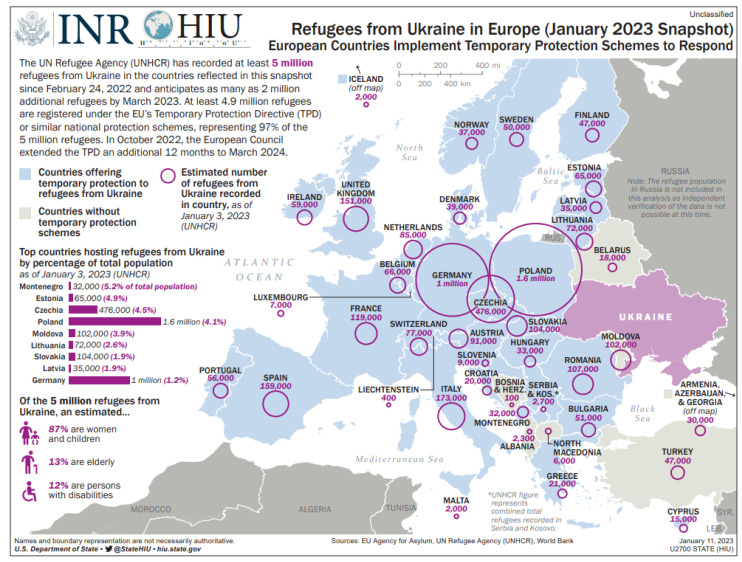
Refugees from Ukraine in Europe. Source: Refugees from Ukraine in Europe (January 2023 Snapshot)—European Countries Implement Temporary Protection Schemes to Respond https://reliefweb.int/report/world/refugees-ukraine-europe-january-2023-snapshot-european-countries-implement-temporary-protection-schemes-respond, (accessed on 13 Jan 2023).

**Figure 3 jcm-12-06554-f003:**
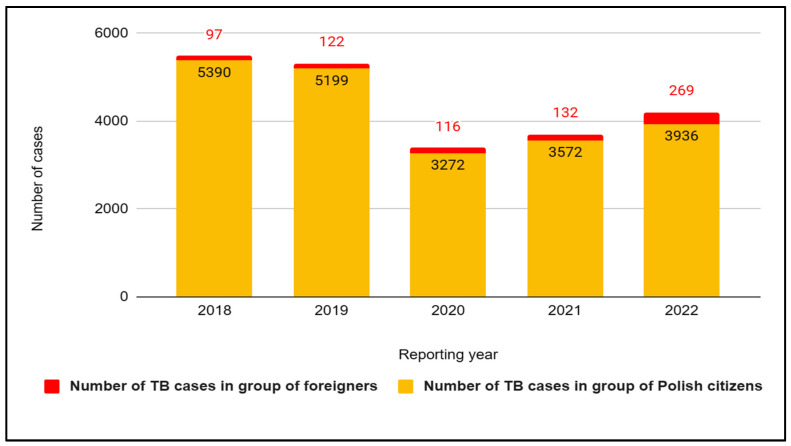
Number of tuberculosis (TB) cases in Poland in groups of Polish citizens and foreigners in 2018–2022.

**Figure 4 jcm-12-06554-f004:**
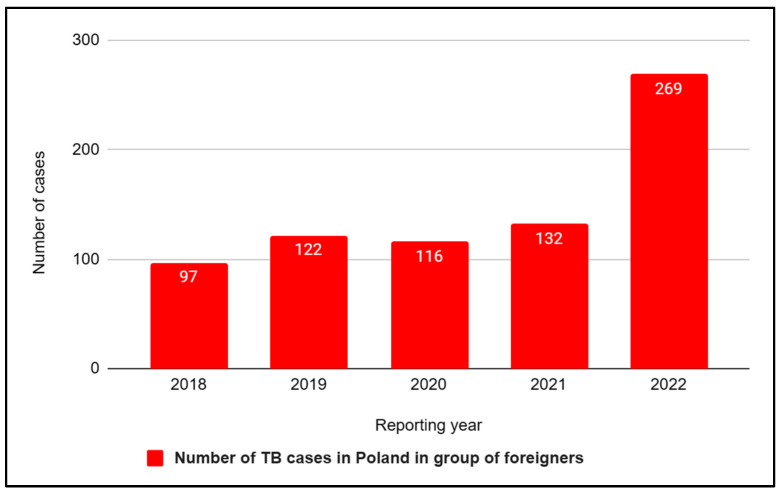
Number of tuberculosis (TB) cases in Poland in group of foreigners in 2018–2022.

**Figure 5 jcm-12-06554-f005:**
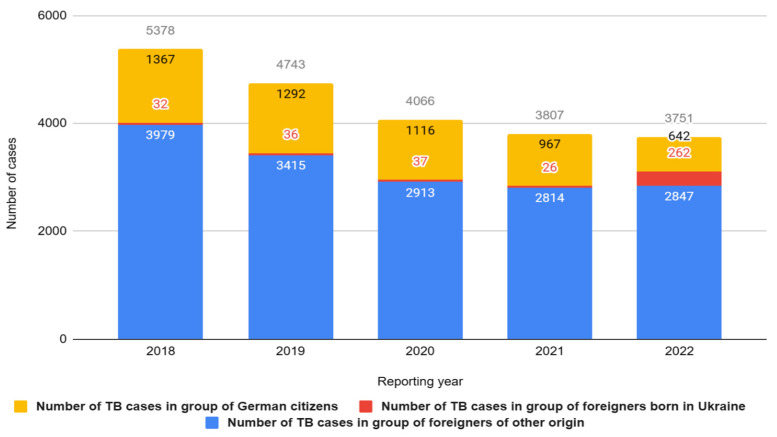
Number of tuberculosis (TB) cases in Germany in groups of German citizens and foreigners, including those born in Ukraine in 2018–2022.

**Figure 6 jcm-12-06554-f006:**
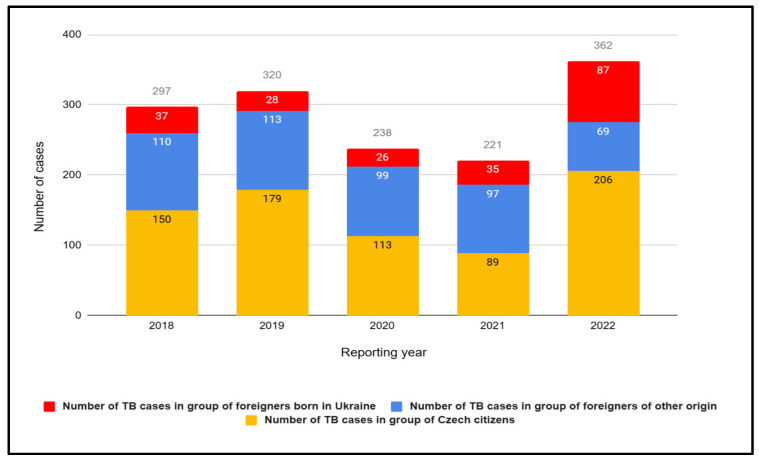
Number of tuberculosis (TB) cases in the Czech Republic in groups of Czech citizens and foreigners, including those born in Ukraine in 2018–2022.

**Figure 7 jcm-12-06554-f007:**
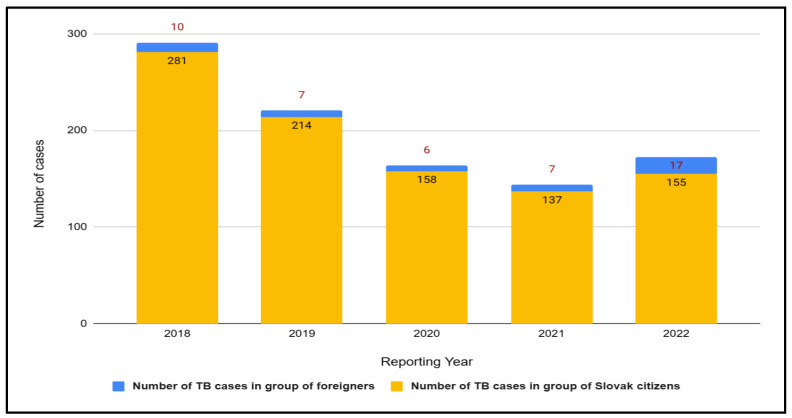
Number of tuberculosis (TB) cases in Slovakia in groups of Slovak citizens and foreigners in 2018–2022.

**Figure 8 jcm-12-06554-f008:**
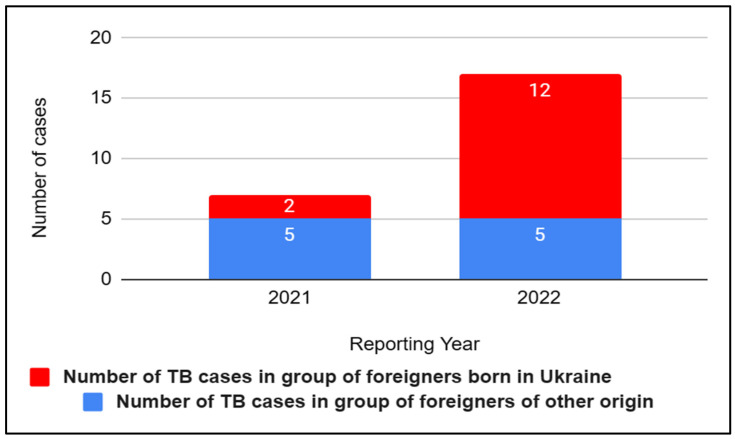
Comparison of number of tuberculosis (TB) cases in Slovakia in groups of foreigners born in Ukraine and foreigners of other origin in 2021–2022.

## Data Availability

Not applicable.

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
