# Peer review of "The Impact of the War in Ukraine on the Epidemiological Situation of Tuberculosis in Europe"

_jcm, 2023, doi:10.3390/jcm12206554_

Round 1
Reviewer 1 Report (Previous Reviewer 1)
Having a chance to review the previous version of this manuscript, I can say that the new version has been significantly improved. I would like to congratulate the authors for improving the manuscript.
The authors might consider the following comments for improving the manuscript
Abstract is fine
introduction
The aim/objective of the study should be added
Results
Figures: I would suggest not using the 3D design for the figures
Discussion
The limitation and strength of the study is missed
Overall well done and congratulations
Author Response
Please see the attachment.

Reviewer 2 Report (Previous Reviewer 2)
The manuscript sheds light on the pressing issue of the rise in tuberculosis (TB) cases in Ukraine, particularly in the ongoing armed conflict and the subsequent migration patterns. While the topic is undeniably important, especially for public health professionals and policymakers, the paper falls short in several areas.
Strengths:
- The paper addresses a timely and significant issue, highlighting the interplay between socio-political upheavals and health crises.
- The authors have made an effort to not only focus on the direct repercussions of the conflict but also to bring to light other associated risk factors such as malnourishment, HIV infections, and vitamin D3 deficiencies.
Major Concerns:
- The manuscript frequently alludes to a significant increase in TB cases but fails to provide specific figures, percentages, or quantitative data. This omission hampers the reader's ability to comprehend the magnitude and severity of the issue fully.
- While the paper mentions factors like HIV infections and malnourishment, it does not delve deep into understanding how these factors are intertwined with the primary issue of conflict and migration.
- The manuscript concludes with a generic call for increased awareness and research. However, it does not offer concrete interventions or strategies that could be implemented to address the crisis.
- The Data collection section is unacceptable.
Recommendations for the Authors:
- The manuscript would be significantly enhanced by including concrete statistics, charts, graphs, and tables that depict the rise in TB cases.
- A more thorough examination of the associated factors and their relationship with the primary issue is crucial. This could involve exploring how the conflict has impacted malnourishment rates or the spread of HIV infections.
- The paper should move beyond a mere call for awareness and suggest specific public health campaigns, strategies to strengthen healthcare infrastructure, and collaborative efforts to address the crisis.
- Consider discussing the long-term implications of the TB epidemic on Ukraine's healthcare system and its neighboring countries.
- Describe the data used in the manuscript in detail.
While the manuscript touches upon a critical issue, it requires substantial revisions to meet the standards of academic rigor and comprehensiveness. The authors are encouraged to address the concerns raised in this review.
Round 2
Reviewer 2 Report (Previous Reviewer 2)
Unfortunately, the authors have not addressed reviewer's comments and recommendations.
This manuscript is a resubmission of an earlier submission. The following is a list of the peer review reports and author responses from that submission.
Round 1
Reviewer 1 Report
This review initiated by Natalia et all is very interesting. There is a need to show how recent crises including the war affect the epidemiology of tuberculosis in Europe or worldwide
Introduction
The introduction is well-detailed, however, the link between war and tuberculosis epidemiology needs to be provided and the rationale for performing this review added
I would suggest to the authors add, the research question or objective of this review in the introduction
Method
Please describe clearly the kind of review you aimed for (systematic? Scopus? Narrative?)
Line 120-129 there are many statements without the references
Major comment: The research question looks very interesting, however, the author this not show the link between epidemiology and the war.
There were very interesting summaries of the general epidemiology of Tuberculosis worldwide, as well its risk factors. All of these risk factors are well summarised already in many previous reviews. I cannot see this review so much innovative
Reviewer 2 Report
The paper provides an essential glimpse into the complex interplay between socio-political upheavals, such as armed conflicts and increased migration, and rising tuberculosis (TB) incidence in Ukraine. Utilizing a selective review methodology, the authors highlight other comorbidities and risk factors that further exacerbate the situation.
Strengths of the Paper:
1. The paper addresses a highly pertinent and topical issue. Understanding the factors contributing to the rise in TB cases can have important implications for public health policies and intervention strategies.
2. The study is commendable for not only focusing on the direct repercussions of the conflict but also elucidating other factors like HIV infections, malnourishment, and vitamin D3 deficiencies.
Drawbacks:
1. Although the paper mentions a significant increase in TB cases, specific figures, percentages, or quantitative data are conspicuously absent. This makes it hard for readers to grasp the magnitude of the problem.
2. The study touches upon associated factors like HIV infections, malnourishment, etc. It would have been beneficial if the paper delved deeper into how these factors intertwine with the primary issue of conflict and migration.
Recommendations for Improvement:
1. The paper would greatly benefit from incorporating concrete statistics. Charts, graphs, and tables can be instrumental in conveying the severity of the TB epidemic in the region.
2. The paper should provide a more in-depth examination of how the listed associated factors interplay with the central theme. For instance, how has the armed conflict possibly exacerbated malnourishment or increased the prevalence of HIV infections?
3. Beyond just a call for increased awareness, the paper should suggest concrete interventions. This might include recommendations for:
- Specific public health campaigns targeting high-risk groups.
- Strengthening healthcare infrastructure in conflict and post-conflict zones.
- Collaborative efforts between governments, NGOs, and international bodies to address the issue.
4. The paper could also touch upon the implications of such a rise in TB for neighboring countries or Ukraine's healthcare system in the long run.
The paper provides an essential analysis of the rise in TB morbidity in the backdrop of the Russian military invasion of Ukraine. While the subject is of great significance, the paper can benefit from a more detailed methodology, incorporation of quantitative data, and actionable recommendations. With these modifications, the study has the potential to offer valuable insights and guide interventions to combat the TB epidemic in the region.
Reviewer 3 Report
Thank you for asking me to review the review manuscript titled: "Tuberculosis in the Face of War. Epidemiology and Risk Factors of Tuberculosis". There are aa few major issues with the manuscript:
1. Major issues with this review manuscript is that the title is misleading to suggest that the authors intend to describe the impact of the War in Ukraine as it may have affected the TB epidemiology in the same country. The authors describe known global epidemiology mainly extracted from the WHO Global TB Report, known risk factors, transmission and occurrence not specific to Ukraine's war
2. In the introduction, the nature of the research question to be answered is not clearly stated. This is implied in the first sentence of the method and materials section, line 71. The objectives of the study is also not given in the introduction.
3. The authors did not include literature on the impact of war on transmission of TB in general inn the introduction. This could then should have been linked with the results and discussion sections.
4. The objective on line 71 suggests that the authors are describing the epidemiology of TB in Ukraine in the face of the war. The results fail to demonstrate the effect of the war inn Ukraine to the burden of TB.
5. The methods and materials section states that "selective review" in line 73. The authors should describe the selective nature of the review to make it clearer to the reader. The authors indicate the sources of data/articles, but did not show the search strategy and Boolen logic.
6. The results section describe the epidemiology, transmission and treatment of TB in general with no specific reference to how the war had affected the same.
7. The burden of TB is presented with reference to different time zones, 2019-2022, Feb to Aug 2022.
8. The flow of the manuscript under results is affected but he presentation of Drug resistant TB and susceptible TB burden ad hoc. The authors could present susceptible TB burden as affected by the war and drug resistant TB as affected by the war under separate paragraphs to improve flow of ideas.
9. The Ukraine war is mentioned in line 138, but there are ni TB or drug resistant TB data included in the description.
10. Treatment outcome of drug resistant TB is quoted as 50% due to the Ukraine war. Authors are encouraged to present treatment outcomes for the period before the war
11. Linen 152 is speculative
13. Risk factors description was presented as general risk factors. Authors could present these in relation to the war. How has the war worked the risk factors. An attempt is done on socio-economic effects and risk of TB.
14. Under discussion, lines 464-468, recommendations have been recommended by WHO for countries to adapt. Are the authors implying these were not yet in the treatment guidelines of Ukraine. This must be stated clearly if this is the caase
Generally the authors are encouraged to seek services of an English editor. The few errors are on lines 442 and 482
Round 2
Reviewer 1 Report
Thanks for providing the response to my comments